# Effects of Immobilization and Swimming on the Progression of Osteoarthritis in Mice

**DOI:** 10.3390/ijms24010535

**Published:** 2022-12-28

**Authors:** Tong Xue, Kaiting Ning, Baoqiang Yang, Xiangya Dou, Shuaiting Liu, Dongen Wang, Huiyun Xu

**Affiliations:** Key Laboratory for Space Bioscience and Biotechnology, School of Life Sciences, Northwestern Polytechnical University, Xi’an 710072, China

**Keywords:** osteoarthritis, DMM, swimming, immobilization

## Abstract

Osteoarthritis (OA) is a chronic joint disease characterized by the degeneration of articular cartilage and thickening and sclerosis of the subchondral bone. Mechanical factors play significant roles in the development and progression of OA, but it is still controversial whether exercise or rest is a more effective treatment for OA patients. In this study, we compared the effects of swimming and immobilization at different stages of OA in mice. Four weeks (the middle stage of OA) or eight weeks (the late stage of OA) after DMM (destabilization of the medial meniscus) surgery, the mice were subjected to four-week immobilization or swimming. Ink blot analysis and a beam walking test were performed to measure the gait and balance ability. Histological analysis was performed to determine the trabecular bone area, the thickness of subchondral bone, the thickness of the cartilage, the OARSI score, and the expression of MMP13 (matrix metalloproteinases) and IL-6 (interleukin). The results showed that at the middle stage of OA, both immobilization and swimming slowed down the progression of OA. Immobilization relieved OA to a certain extent by decreasing the production of regulatory factors to attenuate the degeneration of cartilage, which partly relieved the effects of DMM on gait, mainly in the hindlimb. Swimming mainly attenuated the thickening and rescued the area of subchondral bone.

## 1. Introduction

It is estimated that about 10% of the population over 60 years old have serious clinical problems caused by osteoarthritis (OA) [1]. The pathological features of osteoarthritis mainly include cartilage damage on the joint surface, vascular invasion, thickening and sclerosis of subchondral bone, osteophyte formation, and synovial inflammation, etc. [2,3], which eventually lead to joint stiffness, swelling, pain, and loss of function.

Currently, there are still no effective pharmacological agents for preventing or treating OA [4]. Cartilage tissue engineering and stem cell therapy have raised great expectations for cartilage regeneration and OA treatment yet have not been widely proven to work well [5,6,7,8]. Total joint replacement is still the ultimate choice for OA patients [9]. Therefore, some nonoperative treatments have been used to alleviate the symptoms, reduce pain, and improve life quality [10]. Due to the important role of abnormal mechanical loading in the initiation and progression of OA [11], the effects of exercise on the structure and function of joints have been studied, with some helpful results [12,13,14]. Because high-intensity exercise can cause damage to joints, moderate or low-intensity exercise is recommended to reduce the pain caused by OA, such as swimming [15]. On the other hand, immobilization is also a potential treatment for OA in reducing pain, improving the functionality, and suppressing articular cartilage degeneration [16], especially on carpometacarpal joint OA [17,18].

Therefore, whether exercise or rest is a more effective mechanical treatment for patients with OA, especially regarding the changes in joint structure and function at the middle and late stages of the disease, remains controversial. Some clinical trials demonstrated the positive effects of exercise on OA symptoms in patients [19,20], whereas immobilization was usually used to generate OA models [21] and improve OA [17,18]. In this study, we evaluated the effects of swimming (exercise) and immobilization (unloading) on the progression of OA in male mice. DMM (destabilization of the medial meniscus) surgery was used to generate OA models. Four (middle stage) or eight weeks (late stage) after the DMM surgery, the mice were randomly subjected to swimming or immobilization for another four weeks. For the swimming treatment, the mice were subjected to a swimming procedure for 1 h/day or 40 min/day in different subgroups, 5 days/week. For immobilization, the hindlimb of each mouse was put through a tube and the movement of the knee joint was restricted by the tube. We hope the results can serve as a theoretical basis for future mechanical adjunctive treatment of OA.

## 2. Results

### 2.1. Immobilization Partly Rescues the Effects of DMM on Gait but Aggravates the Disadvantages of DMM on the Balance Ability at the Middle Stage of OA

The ink blot assay showed that 4 weeks after DMM surgery the stride width, step distance, and percentage of right-to-left step length of the forelimbs were not significantly different between DMM and Sham mice (Figure 1A–E,G). Only the stride length and percentage of right-to-left stride length of the hindlimbs were significantly decreased in DMM mice (Figure 1F,H).

The step distance and stride length of the hindlimb were significantly shorter in DMM mice compared with Sham mice at both 8 and 12 weeks after DMM surgery, which was rescued by immobilization but not swimming (Figure 1D,H and Figure 2D,H). In Group M (middle stage of OA, 4 weeks after DMM), 4 weeks of swimming also rescued the decrease in step distance of the hindlimb after DMM (Figure 1D). In contrast, all the parameters were the same between the swimming and DMM mice in Group L (late stage of OA, 8 weeks after DMM) (Figure 2A–H).

The beam walking test showed that the time to walk across the beam and the number of times the mice fell off were significantly increased in DMM than in Sham mice at 4, 8, and 12 weeks after DMM surgery (Figure 3). In Group M, immobilization obviously increased the number of times the mice fell off, whereas swimming decreased the crossing time compared with DMM mice (Figure 3A,B). In Group L, immobilization partly reduced the crossing time (Figure 3C) but not the number of times the mice fell off compared with DMM mice (Figure 3D).

### 2.2. Immobilization and Swimming Both Play Positive Roles in Histopathological Changes in OA at the Middle Stage but Not the Late Stage

Safranin-O/Fast Green staining was used to assess the cartilage morphology and OARSI (Osteoarthritis Research Society International) scoring. Histological images showed a significant cartilage matrix degradation after 8 weeks of DMM compared with Sham mice, and the degradation was more serious after 12 weeks of DMM (Figure 4A). Consistently, at the tibial plateau and femoral condyle, the OARSI score was higher in DMM mice than Sham mice, accompanied by a smaller cartilage area size in both Groups M and L (Figure 4B,C).

As shown in Figure 4B,C, in Group M, histological analysis showed that both immobilization and swimming decreased the OARSI score and increased cartilage area compared with DMM at the tibial plateau but not at the femoral condyle. In Group L, the cartilage area at the tibial plateau was increased only after immobilization, not swimming, compared with DMM.

An immunohistochemical analysis was performed to determine the expression of MMP (matrix metalloproteinases) 13 and IL (interleukin)-6 at the tibial plateau cartilage matrix. As shown in Figure 5, both MMP13- and IL-6-positive cells were significantly increased in DMM compared with Sham mice in Groups M and L. In Group M, immobilization decreased the positive cells of MMP13 and IL-6 compared with DMM. In contrast, swimming only decreased IL-6- not MMP13-positive chondrocytes. In Group L, neither immobilization nor swimming affected the expression of MMP13 and IL-6.

These results showed that DMM significantly caused some phenotypic changes similar to OA, such as a decreased cartilage area, an increased OARSI score, and the expression of matrix regulatory molecules at both the middle and late stages. Both immobilization and swimming increased the cartilage area and attenuated the OARSI score at the middle stage. However, these positive effects were not obvious at the late stage.

### 2.3. Swimming Slightly Relieves the Alteration in Subchondral Bone Only at the Middle Stage of OA, Not the Late Stage

Safranin-O/Fast Green staining was used to measure the thickness of the subchondral bone structure at the medial tibial plateau. A thicker subchondral bone plate was observed in DMM than in Sham mice in both Groups M and L (Figure 6A). In Group M, swimming decreased the thickness of the subchondral bone plate, whereas immobilization did not. In Group L, there was no obvious difference between the three subgroups (Figure 6A).

Moreover, the cross-sectional area of bone trabecula at the tibial metaphysis was smaller in DMM mice than in Sham mice in both Groups M and L. In Group M, swimming rescued the decrease in area, whereas immobilization did not. In addition, no obvious difference was found among the three subgroups in Group L (Figure 6B,C).

These results showed that DMM significantly affects the subchondral bone structure both at the middle and late stages, and swimming, not immobilization, partly relieved the alteration in subchondral bone only in the middle stage, not the late stage.

## 3. Discussion

In this study, we found that immobilization and swimming both have some benefits in slowing down the progression of OA, mainly at the middle stage. Immobilization mainly attenuated the degeneration of cartilage, whereas swimming mainly improved the thickening and area of subchondral bone.

Due to the limits of medicine and the operation treatment for OA, some nonoperative methods, such as mechanical stimulation, are used to alleviate the symptoms and pain. As moderate stimulation, exercise was thought to be advantageous for OA. Previous studies have exhibited that swimming [22] and intermittent gentle exercise avoided the loss of bone mineral density in the subchondral bone during OA [23]. Interestingly, some unloading treatments also decelerated the OA progression. For example, joint immobilization protected against the degradation of cartilage in the progression of OA in mice [24,25], and hindlimb unloading (HLU) inhibited articular cartilage degeneration in the knee joints of a monosodium iodoacetate-induced OA in rats [16]. Especially on the carpometacarpal joint OA, immobilization with orthosis showed some beneficial effects of pain reduction and functional improvement [17,18].

In patients with knee injuries, a decrease in knee flexion (stiffening knee pattern) is commonly observed [26]. A similar symptom was also observed in rats with antigen-induced OA [27]. These pathological changes may affect joint behavior. Here, two assessment methods, the ink blot assay and beam walking test, were used to determine the balance ability in the mice. We found that the change in gait caused by DMM was only in the step distance and stride length of the hindlimb, which is similar to the previous experiment [28]. This may be related to the DMM surgery only being performed on the hindlimb. Immobilization partly rescued the effects of DMM on gait but aggravated the disadvantages of DMM on the balance ability. Since the balance ability was mainly related to joint flexibility, the balance ability may be exacerbated by immobilization. However, the slight improvement caused by immobilization at the late stage of OA may be due to the remission of the cartilage surface area reduction, which partly improves the balance ability. Further studies are needed to verify this.

In this study, immobilization and swimming both slowed the progression of OA at the middle stage, shown as a decreased OARSI score and less overall cartilage degeneration, the two most important variables to evaluate OA progression. Previous studies have also shown that joint immobilization or hindlimb unloading prevented the occurrence of OA, accompanied by a decreased OARSI score [16,24]. In addition, 7 days of hindlimb unloading following ACL joint injury reduced inflammation, preserved bone volume, and potentially slowed OA progression, though these were not maintained at later time points [29]. Conversely, long-term immobilization has a degenerative effect on the joint, leading to OA-like changes even in uninjured limbs [30,31]. Takahashi et al. indicated that hindlimb unloading increased OA progression in mice, with higher OARSI scores and thinner articular cartilage [32]. These conflicting results are most likely related to different time points of unloading treatment and detection. Further studies with more time points are required for investigating the effects and mechanism of unloading on OA progression. Moreover, immobilization showed a region-specific effect on cartilage degeneration, mainly at the tibial plateau but not the femoral condyle. That may be because immobilization kept the hindlimb of the mice in full extension, potentially changing the mechanical stress on the tibial plateau and femoral condyle.

Moderate mechanical stress has been shown to benefit OA patients [20]. In mouse models, well-controlled, low-level loading shortly following OA joint trauma attenuated early OA [33,34]. In our study, swimming, as a moderate exercise, improved cartilage degeneration and the OARSI score at the middle stage of OA. That may be related to the role of swimming in the production of IL-6 and subchondral bone volume and sclerosis. In recent years, the thickening, sclerosis, and increased porosity of subchondral bone during OA progression have been identified [35,36]. In our study, the thickness of the subchondral bone plate increased significantly after DMM, along with a decrease in trabecular area, which was consistent with the previous results in DMM rat models [37]. Swimming, not immobilization, improved the recovery of subchondral bone after DMM. Allen et al. have shown that treadmill exercise blocks intra-articular monosodium iodoacetate injection-induced medial subchondral bone loss and trabecular bone loss in the metaphysis [38]. The mechanisms of the positive effects of exercise on OA have not been fully elucidated. Moderate mechanical stress, which improves the activity of osteoblasts and chondrocytes, likely regulates the formation ability of bone during OA [39,40], and a reduction in local inflammation seems to be another potential reason [41]. Further experiments are required to confirm this.

At both the middle and late stages of OA, immobilization partly reduced the improved expression of IL-6 and MMP13 caused by DMM surgery, whereas swimming only reduced the expression of IL-6 at the late stage. IL-6 and MMP13 were thought to be important regulatory factors to initiate the degradation of cartilage matrix and subchondral bone and promote OA progression [42,43]. A decrease in IL-6 and MMP13 causes reduced cartilage catabolism, matrix degradation, and resorption. That may be how immobilization and swimming protect against a reduction in the cartilage area caused by DMM.

Recently, a molecular classification of OA has been suggested according to its different subtypes [44]. Personalized prevention and treatment strategies for OA patients have become necessary [45]. Therefore, different mechanical adjuvant therapies for different OA subtype patients may be a direction for future personalized medicine in OA.

Moreover, only male rats were used in this study, but it is worth noting that a sexually dimorphic phenomenon was found in some previous studies [46]. Would the response to swimming and immobilization be different in female mice? What is the exact cellular mechanism behind the phenomenon caused by swimming and immobilization after OA surgery? How do the cells in articular cartilage and subchondral bone respond to different mechanical environments? Further investigations are required.

## 4. Materials and Methods

### 4.1. Experimental Animals and DMM Operation

The C57BL/6J male mice were obtained from the Animal Center of Air Force Medical University. They were housed on a 12 hr light/dark cycle at 25 °C with 40% relative air humidity and free access to water and feed and were permitted unrestricted activity in a SPF laboratory animal room for at least 1 week. Then, 3-month-old mice (body weight: 20–25 g) were divided randomly into DMM and Sham groups. DMM surgery was performed to induce OA on the right knee joint of 3-month-old mice as previously described [47]. After intraperitoneal injection with sodium pentobarbital (4.5 mg/100 g body weight), the joint capsule of the right knee was exposed under a microscope and the medial meniscotibial ligament was transected. A sham operation was performed using the same approach without ligament transection. It is thought that after 4 weeks of DMM the symptoms of the joint are similar to the middle stage of human OA, whereas 8 weeks is similar to the late stage [48], so these two time points were chosen for the next experiments to evaluate the effects of swimming and immobilization. All experiments were conducted in the Animal Research Lab of Northwestern Polytechnical University (NPU), and all animal protocols were approved by the NPU Institutional Animal Care and Use Committee.

### 4.2. Immobilization and Swimming Treatment

As outlined in Figure 7A, after the DMM operation, the mice were randomly divided into two groups: Group M (middle stage of OA, 4 weeks after DMM) and Group L (late stage of OA, 8 weeks after DMM). Each Group (M or L) was divided into four subgroups, including Sham, DMM, immobilization (IM), and swimming (Swim) for the next four weeks. The mice in Group M were subjected to immobilization or swimming from 4 to 8 weeks after DMM, whereas Group L was from 8 to 12 weeks after DMM.

Immobilization was carried out according to a previous protocol with minor revision [49]. A 1.5 mL centrifuge tube with the bottom cut off was fixed to one end of a clip with foam tape. The hindlimb of the anesthetized mouse was put through the tube and the other end of the clip was fixed to the ankle with foam tape. Thus, the movement of the knee joint was restricted by the tube. Mice in the swimming subgroup were subjected to a swimming procedure in 37 ± 2 °C water for 4 weeks, 5 days/week, according to the previous protocol with minor modification [50], 1 h/day (Group M) or 40 min/day (Group L). After that, all mice were sacrificed (Figure 7B).

### 4.3. Ink Blot Analysis

To assess the balance ability of the mice, ink blot analysis was performed twice the day before immobilization/swimming and the day before sacrifice (Figure 7C) using a tunnel named the CatWalk system, according to a previous protocol [51]. The front paws of the mice were dyed with red ink and the hind paws with blue ink. Pictures of the footprints were taken and then analyzed with ImageJ. The static paw parameters listed in Table 1 were measured and calculated.

### 4.4. Beam Walking Test

The day before immobilization/swimming and the day before sacrifice, the beam walking test was carried out according to a previous protocol [52] (Figure 7D). A lightproof black cubic box with a length, width, and height of 20 cm was placed on a platform about 60 cm above the ground; one end of a one-meter beam with a 2 cm diameter was placed on the platform with an angle of 15° to the horizontal. A camera was placed at the other end of the beam to record the time taken to walk across the beam and the frequency of falling off the beam.

### 4.5. Histological Analysis

The whole joint was fixed in 4% paraformaldehyde for 48 h and then decalcified in 10% ethylenediaminetetraacetic acid for 28 days. After dehydration and paraffin embedding, 5 μm sections were made according to a previous method [53].

For Safranin-O/Fast Green staining, the sections were stained by Fast Green for 10 min and then Safranin-O for 20 s. After washing with absolute ethyl alcohol, the sections were dehydrated and sealed. The sections were observed under a microscope (CIC, XSP-C204) and images were taken with a digital camera (Nikon, 80i). The histology was analyzed in two regions of the joint cartilage: medial femoral condyle (MFC) and medial tibial plateau (MTP). Osteoarthritis Research Society International (OARSI) score was assessed at the area of articular surface as described previously [54]. The mean value from two different observers in a double-blind experiment was taken as the final score.

The cross-sectional area of bone trabecula at the tibial metaphysis was measured and analyzed using ImageJ, and the mean thickness of the subchondral bone plate from 6 random areas was also measured using ImageJ. At least three sections were measured in each subgroup.

### 4.6. Immunohistochemistry (IHC)

After dewaxing and antigen retrieval, the sections were washed 3 times with phosphate buffer solution (PBS, pH = 7.4). Then, they were soaked in 0.3% hydrogen peroxide for 25 min to block endogenous peroxidase at room temperature and washed on a decolorizing shaker three times. After 30 min incubation with 3% Bovine Serum Albumin (BSA), the primary antibody against MMP13 (Abcam, 1:100) or IL-6 (Servicebio, 1:100) in PBS was incubated overnight at 4 °C. The secondary antibody was incubated for 50 mins followed by diaminobenzidine solution (DAB). Nuclei were counterstained with hematoxylin for 3–8 min. Then, the sections were dehydrated and sealed and analyzed using ImageJ.

### 4.7. Statistical Analysis

All experiments were repeated at least three times. The normality test was analyzed with the Shapiro–Wilk test using SPSS v.13.0 software (SPSS Inc., Chicago, IL, USA). Statistical analysis was performed using GraphPad Prism version 5 (La Jolla, CA, USA). All the data were shown as mean ± standard error (SE). *p* value < 0.05 was considered statistically significant. The difference between Sham and DMM was evaluated using t-test, and the difference among the three subgroups, DMM, immobilization, and swimming, was evaluated using one-way ANOVA with Tukey test. “*n*” represents the number of independent observations of different animals in each subgroup.

## 5. Conclusions

In short, this study provides experimental verification of the rehabilitation of OA. Immobilization and swimming both slow down the progression of OA, mainly at the middle stage. Immobilization relieves OA to a certain extent, maybe via decreasing the production of regulatory factors to attenuate the degeneration of cartilage, which partly relieves the effects of DMM on gait, mainly in the hindlimb. Swimming mainly improves the thickening and area of subchondral bone.

These results suggest the potential utilization of mechanical treatments in the rehabilitation therapy of OA patients mainly at the middle stage of OA. The limitations of our study are that only male rats were used and the cellular mechanism of the mechanical treatments for OA progression was still unclear. Future studies are needed to fully determine the influence of mechanical factors on OA progression.

## Figures and Tables

**Figure 1 ijms-24-00535-f001:**
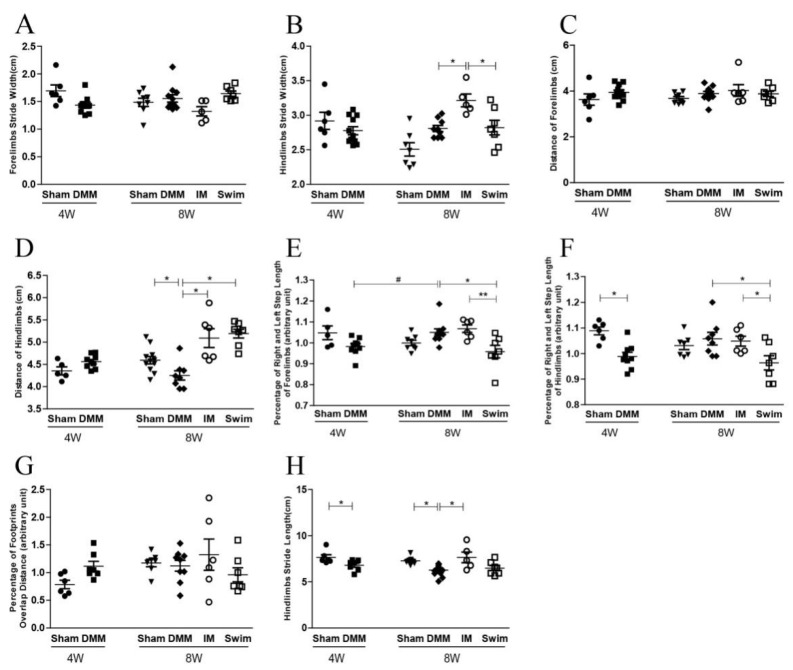
Ink blot analysis showing the effects of swimming and immobilization on the progression of OA at the middle stage. (**A**) Forelimb stride width; (**B**) hindlimb stride width; (**C**) distance of forelimbs; (**D**) distance of hindlimbs; (**E**) percentage of right and left step length of forelimbs; (**F**) percentage of right and left step length of hindlimbs; (**G**) percentage of footprint overlap distance; (**H**) hindlimb stride length. 4W: the time point of starting swimming or immobilization, which is 4 weeks after DMM surgery. 8W: the time point after 4 weeks of swimming and immobilization, which is 8 weeks after DMM surgery. IM, immobilization; DMM, destabilization of the medial meniscus; * *p* < 0.05, ** *p* < 0.01, ^#^
*p*<0.05 (difference between 4W and 8W after DMM), *n* ≥ 6.

**Figure 2 ijms-24-00535-f002:**
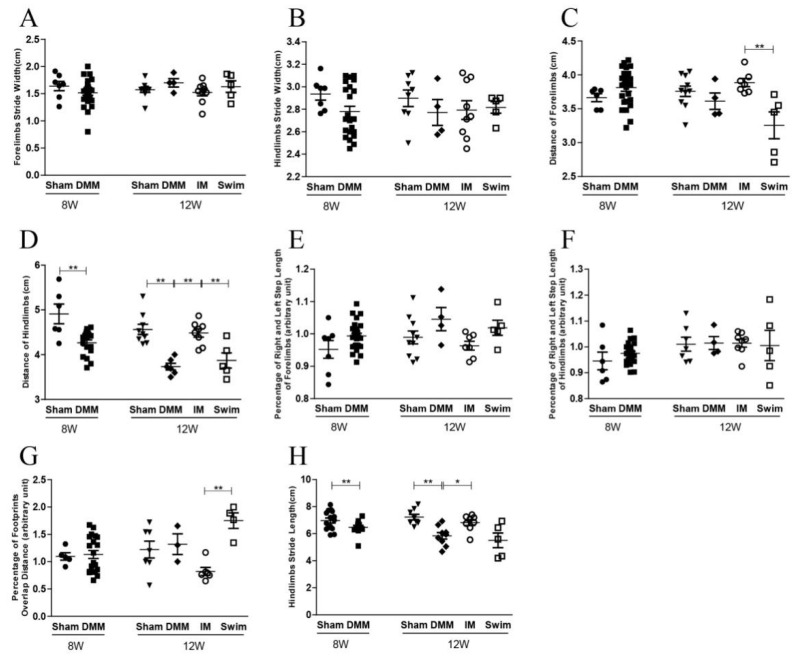
Ink blot analysis showing the effects of swimming and immobilization on the progression of OA at the late stage. (**A**) Forelimb stride width; (**B**) hindlimb stride width; (**C**) distance of forelimbs; (**D**) distance of hindlimbs; (**E**) percentage of right and left step length of forelimbs; (**F**) percentage of right and left step length of hindlimbs; (**G**) percentage of footprint overlap distance; (**H**) hindlimb stride length. 8W: the time point of starting swimming or immobilization, which is 8 weeks after DMM surgery. 12W: the time point after 4 weeks of swimming and immobilization, which is 12 weeks after DMM surgery. IM, immobilization; DMM, destabilization of the medial meniscus; * *p* < 0.05, ** *p* < 0.01, *n* ≥ 6.

**Figure 3 ijms-24-00535-f003:**
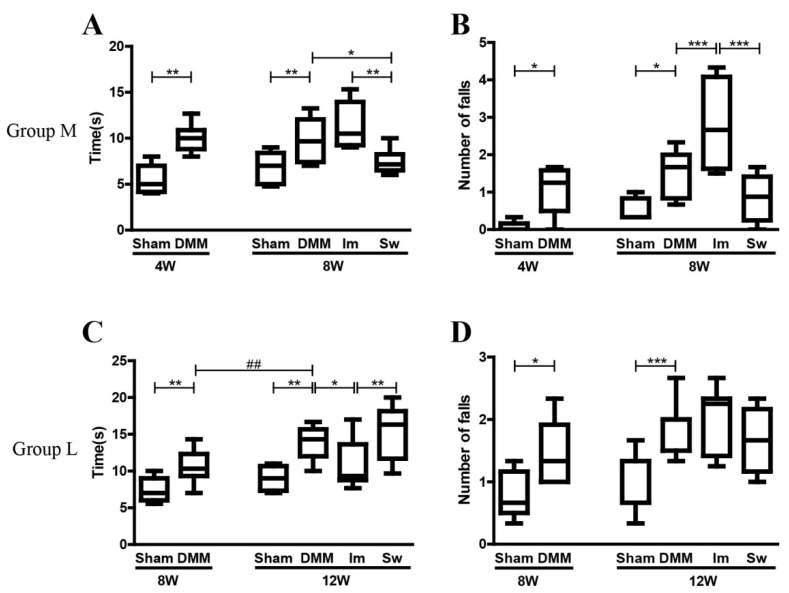
Beam walking test showing that immobilization aggravated the disadvantages of DMM on the balance ability at the middle stage of OA. (**A**) DMM significantly increased the time to cross the beam and (**B**) the number of times the mice fell off the beam, and immobilization further aggravated the imbalance, whereas swimming reduced the time at the middle stage of OA. (**C**) Immobilization reduced the time to cross the beam (**D**) but not the number of times the mice fell off the beam, whereas swimming had no effects at the late stage of OA. Group M: middle stage of OA, 4 weeks after DMM and Group L: late stage of OA, 8 weeks after DMM. 4W: the time point of starting swimming or immobilization, which is 4 weeks after DMM surgery. 8W (**A**,**B**): the time point after 4 weeks of swimming and immobilization, which is 8 weeks after DMM surgery. 8W (**C**,**D**): the time point of starting swimming or immobilization, which is 8 weeks after DMM surgery. 12W: the time point after 4 weeks of swimming and immobilization, which is 12 weeks after DMM surgery. Im, immobilization; Sw, swim; DMM, destabilization of the medial meniscus; * *p* < 0.05, ** *p* < 0.01, *** *p* < 0.001, ^##^ *p*<0.01 (difference between 8W and 12W after DMM), *n* ≥ 6.

**Figure 4 ijms-24-00535-f004:**
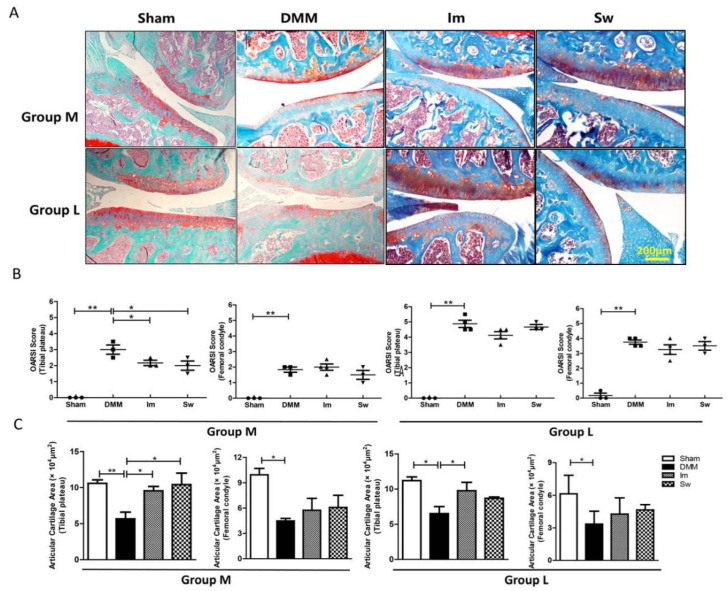
Safranin-O/Fast Green staining showing that immobilization and swimming both play positive roles in the cartilage area and OARSI score, mainly at the middle stage. (**A**) Safranin-O/Fast Green staining of the cartilage (red) and subchondral bone (green). (**B**) DMM significantly increased the OARSI score at both the tibial plateau and femoral condyle after 8 and 12 weeks of DMM. Immobilization and swimming both decreased the OARSI score at the tibial plateau in Group M but not Group L. (**C**) Immobilization rescued the decrease in articular cartilage area at tibial plateau in both groups after 8 and 12 weeks of DMM, whereas swimming only increased the area at tibial plateau in Group M. Group M: middle stage of OA, 4 weeks after DMM and Group L: late stage of OA, 8 weeks after DMM. Im, immobilization; Sw, swim; DMM, destabilization of the medial meniscus; * *p* < 0.05, ** *p* < 0.01, *n* ≥ 3; bar = 200 μm.

**Figure 5 ijms-24-00535-f005:**
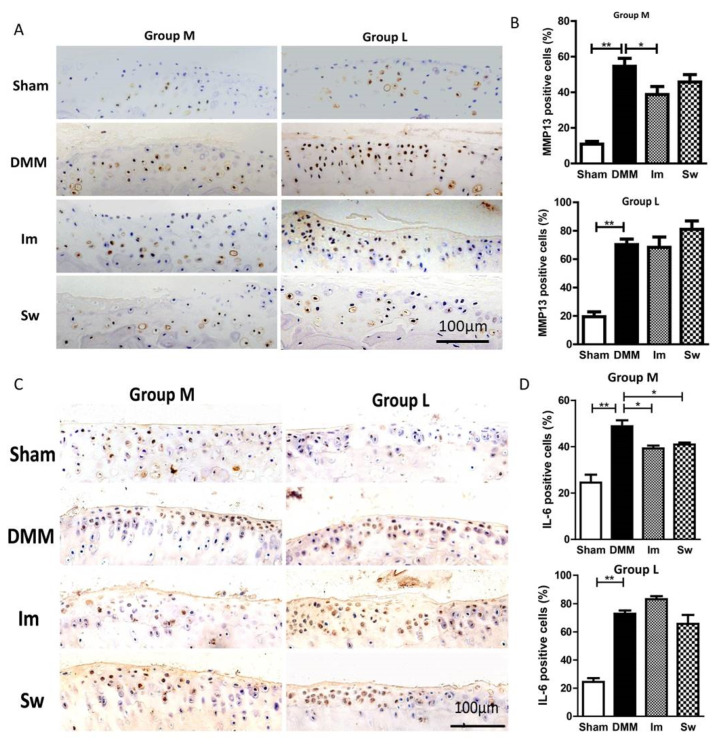
Immunohistochemistry showing the MMP13- and IL-6-positive cells in the chondrocytes of the knee joint. (**A**) MMP13- and (**C**) IL-6-positive cells. (**B**,**D**) In Group M, immobilization partly reduced the increase in MMP13- and IL-6-positive cells caused by DMM, whereas swimming only decreased IL-6 expression. Group M: middle stage of OA, 4 weeks after DMM and Group L: late stage of OA, 8 weeks after DMM. Im, immobilization; Sw, swim; DMM, destabilization of the medial meniscus; * *p* < 0.05, ** *p* < 0.01, *n* ≥ 6; bar = 100 μm.

**Figure 6 ijms-24-00535-f006:**
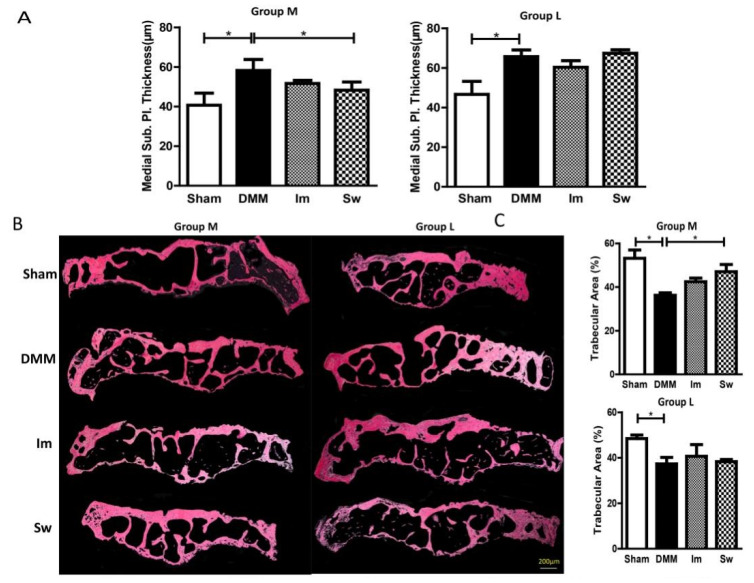
Swimming partly relieved the increase in thickness and the decrease in trabecular area in subchondral bone in Group M. (**B**) The morphology of subchondral bone. (**A**) DMM significantly increased the thickness and (**C**) decreased the area of subchondral bone in both Groups M and L. Swimming partly attenuated the alteration caused by DMM in Group M. Group M: middle stage of OA, 4 weeks after DMM and Group L: late stage of OA, 8 weeks after DMM. Im, immobilization; Sw, swim; DMM, destabilization of the medial meniscus; * *p* < 0.05, *n* ≥ 6; bar = 200 μm.

**Figure 7 ijms-24-00535-f007:**
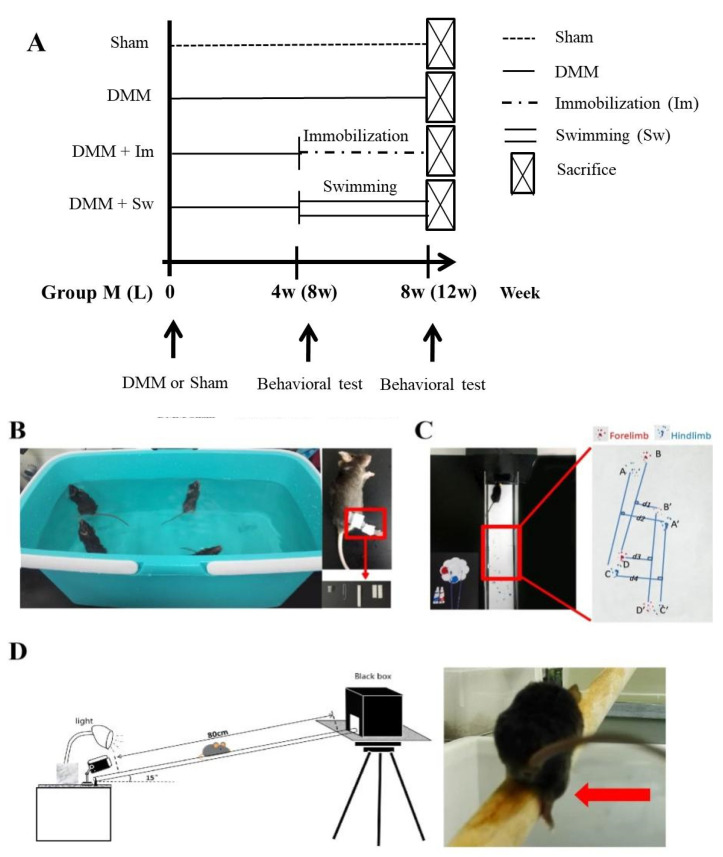
Diagrams and photos of mice grouping and experimental methods. (**A**) Diagram of mice grouping. (**B**) Swimming and immobilization treatment. (**C**) Photo of ink blot test and the footprints of the mice; their front paws were dyed with red ink and the hind paws with blue ink. (**D**) Diagram of beam walking test. The red arrow shows the moment of mice falling off the beam. Group M: middle stage of OA, 4 weeks after DMM and Group L: late stage of OA, 8 weeks after DMM. DMM, destabilization of the medial meniscus.

**Table 1 ijms-24-00535-t001:** Explanation of CatWalk parameters.

CatWalk Parameter	Explanation
Stride length	Distance between the placement of a paw and the subsequent placement of the same paw, BD and B’D’ (Forelimb, F) or AC and A’C’ (Hindlimb, H)
Stride width	Vertical distance from the paw to the midperpendicular, (d1 + d3)/2 (F) or (d2 + d4)/2 (H)
Step distance	Distance between several continuous imprints, (BB’ + B’D + DD’)/2 (F) or (AA’ + A’C + CC’)/2 (H)
Percentage of right-to-left stride length	Percentage of average stride length of right to left paw, B’D’/BD × 100% (F) or A’C’/AC × 100% (H). If it was >1, right stride length was longer than left

## Data Availability

Data sharing not applicable.

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
