# Peer review of "Effects of Immobilization and Swimming on the Progression of Osteoarthritis in Mice"

_ijms, 2022, doi:10.3390/ijms24010535_

Round 1

Reviewer 1 Report

The main concens of this study is "Study design". In clincal practice, the joint immbolization has been treated at early stage of rehabilitation, but in this study authors chose the late stage of rehabilitation. This may seriously affects the results and misleading interpretion of the conclusion. Also, swimming does not belong to loading, but treadmill exercise does. 

Other commenst:

1. Introduction. Rationale is not so clear. Immboilization animal study have been extensively investiaged in osteoarthirst model. Alos, weight-bearing (e.g, treamill) or non-weight bearing (unloading exerise, eg. continunous passive motion) in small and large animals have been studies. Authors should review more and point out why compare with swimming and immbolization. Espeically, why immboilize at the late stage of healing stage. 

2. The animal study wrtting format is suggested to follow "The ARRIVE guidelines (Animal Research: Reporting of In Vivo Experiments)". And, how to determine the sample size? 

3. How to control the exercise dosage of each mice in swimming bucket? Please provide the detailed information. 

4.  Please clarify the relability of histological scoring (Intraclass Correlation Coefficient ). How many  assessors? Blined assessment? and how about the inter-rater ICC. 

5.  Figure 5. This images shown were not correlated to the bar charts. Please confirm. 

6.  Statiscital analysis. Please clarify the normality test in the content.  

Author Response

Thank you very much for offering us such valuable and constructive comments to improve our manuscrip. After reading the reviewers’ comments carefully, we corrected the grammar and modified the manuscript according to the suggestion one by one, and listed our answers to the questions point to point as following. 

Reviewer 2 Report

Thank you for opportunity to review the article " Effects of Immobilization and Swimming on the Procession of Osteoarthritis in Mice”.

TITLE

-       Could you add the type of study in the title?.

ABSTRACT

-       Method. Could you add the type of study?.

INTRODUCTION and discussion

-       Could you consider this recent document about OA?:

a. Osteoarthritis: a call for research on central pain mechanism and personalized prevention strategies. 

 METHODS.

Ok .

RESULT

Ok

DISCUSSION

First paragraph. Could you be present the result in brief.

Could you consider these recent documents about immobilization in OA?:

a. Effect of immobilization of metacarpophalangeal joint in thumb carpometacarpal osteoarthritis on pain and function. A quasi-experimental trial.

b. Necessity of Immobilizing the Metacarpophalangeal Joint in Carpometacarpal Osteoarthritis: Short-term Effect.

Tables

ok

Author Response

(The authors gave the same response as above.)

Reviewer 3 Report

comments to the Authors - 

 Regarding the article titled "Study of Immobilization and Swimming on the Procession of Osteoarthritis in Mice".

There are the following prominent opinions :

 Overall, this manuscript is an interesting and straightforward concept at the heart of this illustrated manuscript.

  1. This article has serious English grammar issues, particularly with the use of informal language rather than formal writing. Therefore, it is highly recommended that the language quality be improved.
  2. The primary objective of this paper is to examine the effects of immobilization and swimming on osteoarthritis patients using a DMM mouse model. The results of immobilization and exercise on OA progression have been demonstrated previously by clinical studies. However, the focus of this article is the use of swimming in middle-stage and late-stage OA mice. The author must provide more information regarding how swimming and immobilization affect this situation.
  3. Line 46 - 47 should be corrected by removing the question mark and rephrasing it as follows: "Therefore, exercise or stillness, is the more effective mechanical treatment for patients with osteoarthritis, especially in the middle and late stages of the disease"
  4. The author stated that mechanisms should be investigated. Therefore, lines 47-49 should be removed or edited accordingly.
  5. Swimming (exercise) and immobilization methods the author used in this article need to be described in the introduction section.
  6. The author can investigate the Immobilization effect on Osteophyte maturity from Figure 6D 
  7. The author described the pain-reducing effect of immobilization and swimming; however, there was no experiment to prove it here. In this regard, I suggest the use of a pain assay. 
  8. Lines 190- 193 need to shift in the introduction section. 
  9. The author can describe why particularly select MMP13 and IL-6 instead of MMP (1-15) and other pro-inflammatory cytokines. 
  10. In all figures, the "X" axis's caption is confusing. 8W(Sham DMM), 12W (Sham DMM IM Swim). Did IM and Swim group perform DMM surgery? 
  11. What are the differences between group M and group L? 
  12. In figure 5, represented images and quantification graph did not match. 
  13. For further validation and to improve the paper author can add Ink blot assay and Beam walking test image data.
  14. In conclusion, the author explained the limitations of studies that need to be deliberate more in the discussion section.

In addition, there are some minor comments :

  1. Provide DOI links of references and modify the format of the references by referring to other articles in this journal.
  2. There are many apparent errors in the article, some special words need to be italicized, and the first occurrence in the article needs to be spelled out.

Date Sent:

16-11-2022

Round 2

Reviewer 3 Report

Comments 9 and 10 did not respond well. 

Please apply the reviewer's response contents to the Figure legends.  This revised form did not apply to appreciate. 

Author Response

Response: Thanks for your suggestion. We added the contents to the Figure Legends in the revision. Also, we revised Fig.7A to make the experiment design clearer.